# The effects of extended photoperiod and warmth on hair growth in ponies and horses at different times of year

Christiane O'Brien[1], Megan Ruth Darcy-Dunne[2], Barbara Anne Murphy[2]*

**1** Equilume Ltd., Naas, Co. Kildare, Ireland, **2** School of Agriculture and Food Science, University College Dublin, Belfield, Dublin, Ireland

◉ These authors contributed equally to this work.

* barbara.murphy@ucd.ie

**Data Availability Statement:** All relevant data are within the paper and its Supporting Information files.

## Abstract

Photoperiod is considered the most dominant environmental cue allowing animals to anticipate and adapt to seasonal changes. In seasonally breeding mammals, changes in daylength alter pineal melatonin secretion and pituitary prolactin secretion. During the seasonal transition to shorter winter daylengths, increased production of melatonin and declining prolactin are associated with triggering winter coat growth in many animals. Similarly, studies have shown that artificial extension of photoperiod suppresses melatonin secretion and lifts prolactin inhibition to activate moulting. Four longitudinal cohort studies were conducted to determine if extended photoperiod and warmth, provided by mobile light masks and rugs (horse blankets), could reverse the onset of winter coat growth, maintain the summer coat and accelerate winter coat shedding in horses and in ponies. Studies began at dates corresponding to the autumnal equinox, one month post-summer solstice, one month pre-winter solstice and one month post-winter solstice, respectively. To extend photoperiod to approximately 15h of light, commercially available head-worn light masks provided low intensity blue light to one eye until 11pm daily. Coat condition and shedding rate were scored and hair samples collected, measured and weighed bi-weekly. Data from control and treatment groups were analysed by repeated measures ANOVA. Results revealed that extended photoperiod 1) did not reverse winter coat growth when initiated at the autumnal equinox, 2) effectively maintained the summer coat in stabled horses when initiated one month post-summer solstice, 3) accelerated shedding in outdoor living horses when initiated one month pre-winter solstice and 4) did not accelerate shedding in indoor or outdoor living ponies when initiated one month post-winter solstice. To successfully manage equine coat growth while also preserving optimal thermoregulation in both competition and breeding stock correct timing of light application is crucial and requires careful monitoring of environmental temperature. Further studies are needed where variations in breed and management are considered.

**Funding:** The funders Equilume Ltd. [www.equilume.com] and UCD's School of Agriculture and Food Science [www.ucd.ie] provided support in the form of research material and salaries for authors COB and BAM, respectively, but did not have any additional role in the study design, data collection and analysis, decision to publish, or preparation of the manuscript. The specific roles of these authors are articulated in the 'author contributions' section.

**Competing interests:** I have read the journal's policy and the authors of this manuscript have the following competing interests: BAM is the Founder of Equilume Ltd., a spin-out company deriving from her research program as assistant professor at UCD and is a member of the company's Board of Directors. BAM is a shareholder in Equilume Ltd. COB is and employee of Equilume Ltd in the role of Research Manager. The light mask used in the presented studies is a commercially available product with the following patents: AU2012101968 GB2504244 GB2549682 US9,839,791 This does not alter our adherence to PLOS ONE policies on sharing data and materials.

## Introduction

Annual changes in equine pelage are an adaptation to seasonal variation in environmental conditions. Retinal reception of seasonally changing environmental light information (24h light/dark cycles) is transmitted via the retino-hypothalamic tract to the bilaterally paired suprachiasmatic nucleus (SCN) in the hypothalamus [1]. The SCN, also referred to as the 'master clock' or 'pacemaker' of the body, encodes these 'time-of-day' signals and disseminates them throughout the organism to drive circadian and circannual changes in physiology [2]. At the pineal gland, the SCN signal affects melatonin secretion [3] such that daylength duration is inversely proportional to duration of melatonin production, highlighting its central role in regulating mammalian physiological responses to photoperiod [4]. In many species, including horses, melatonin has been found to suppress prolactin secretion [5,6], which has many biological functions including the regulation of the hair follicle cycle [7]. This cycle comprises three main phases: anagen (active growth), catagen (involution) and telogen (quiescence).

Prolactin receptors in follicle cells allow detection of changes in prolactin-signalling and alter the timing and duration of telogen [8]. Therefore, it was proposed that the transition between telogen and anagen and the moulting process are initiated by changes in daylength and the associated reciprocal changes in melatonin and prolactin [9–11].

Secondary to photoperiod, temperature is also considered an important cue for initiation of physiological changes in many seasonal animals [12]. Photoperiod and temperature are thought to interact in driving seasonal adaptations in timing of first ovulation [13] and hair coat [14] in horses.

As the natural daylight spectrum peaks within the blue portion, photoreception involves intrinsically photosensitive retinal ganglion cells (ipRGCs) [15] that are optimally stimulated by short wavelength blue light [16]. Recently, it was demonstrated that short wavelength blue light administered via head-worn light masks to a single eye, effectively suppresses melatonin production in horses [17] and can be used to extend photoperiod, advance the initiation of seasonal reproductive activity [18], influence seasonal regulation of gestation length and foal birth weight and regulate hair coat development before birth [19]. This adds to the body of work showing the efficacy of blue wavelength light at suppressing melatonin in multiple species [20–22].

The growth of a heavier winter coat, characterised by longer, thicker hairs, is undesired by many horse owners due to its impact on optimal thermoregulation in intensely exercised competition animals and visual aesthetics in show animals. Providing an alternative to the time-consuming and laboursome common practice of shaving or 'clipping' a horse's coat could be advantageous. Additionally, the use of artificially extended lighting to manipulate reproductive cycles is common practice within the global equine breeding industry [23]. Therefore, developing a better understanding of how coat growth responds to light has important relevance for all horses in order to optimise thermoregulation for improved health and welfare in both competition and breeding stock. We hypothesise that, similar to other circannual rhythms in physiology that display phase response curves [24], the annual cycle of coat growth will respond differently to an extended daily light stimulus to advance or delay coat shedding or growth in a time-of-year specific manner. This compilation of studies aimed to evaluate if extended photoperiod and warmth, provided in the form of mobile light masks and rugs, could reverse the onset of winter coat growth (study 1), maintain the summer coat (study 2) and accelerate winter coat shedding in horses (study 3) and in ponies (study 4).

## Materials and methods

Four studies were designed to evaluate the effects of extended photoperiod and warmth on equine coat growth at various times of the year. Studies were conducted in accordance with

**Table 1. Group characteristics of horses and ponies at the start of each study.**

| Study | Group | Age | Hair length (mm) | 10-Hair weight (µg) |
|---|---|---|---|---|
| 1 | Treatment | 16.0 ± 5.4 * | 13.6 ± 1.6 | 435.4 ± 195.1 |
|  | Control | 10.0 ± 2.3 * | 16.6 ± 5.0 | 459.0 ± 212.6 |
| 2 | Treatment | 12.2 ± 4.4 | 8.9 ± 1.1 | 254.7 ± 78.0 |
|  | Control | 14.0 ± 5.4 | 10.6 ± 2.3 | 416.9 ± 170.2 |
| 3 | Treatment | 11.8 ± 6.6 | 28.4 ± 4.0 | 770.6 ± 144.2 |
|  | Control | 9.4 ± 4.8 | 25.4 ± 2.7 | 772.3 ± 288.9 |
| 4 | Treatment | 8.6 ± 5.7 | 30.4 ± 1.4 | 1407.6 ± 454.6 |
|  | Control | 9.1 ± 6.4 | 33.4 ± 7.3 | 1306.7 ± 623.1 |

Values are represented as Mean ± SD.

* denotes significance at $P < 0.05$ as determined by two-tailed T-test analysis.

the 'Code of Good Practice In Research' (University College Dublin, Ireland) and 'Directive 2010/63/EU of the European Parliament and of the Council on the Protection of Animals used for Scientific Purposes'. The studies described were exempt from review by University College Dublin's Animal Research Ethics Approval Committee as they met the following criterion for exemption: 'The study does not involve euthanising a living animal or conducting a *procedure* on a living animal', where '*procedure*' is defined as 'any use, invasive or non-invasive, of an animal for experimental or other scientific purposes, with known or unknown outcome, or educational purposes, which may cause the animal a level of pain, suffering, distress or lasting harm equivalent to, or higher than, that caused by the introduction of a needle in accordance with good veterinary practice' in S.I. No. 543/2012—European Union (Protection of Animals used for Scientific Purposes) Regulations 2012.

For each study, all participating animals were at similar stages of coat growth and had a similar photoperiodic history. Animals were randomly allocated to groups after blocking for type (horse/pony), age and gender. Medium weight all-weather rugs (280g polyester fill), suitable for the Irish climate and a low temperature range of -6.7 to 4.4˚C were used and included neck covers for maximum insulation. Natural daylength (sunrise to sunset) was recorded for each study's specific location using an online resource (www.timeanddate.com). Regional temperatures were tracked from the nearest official weather station of the Irish Meteorological Service (www.met.ie). Data on age, hair length and hair weight at the start of each study are summarised in **Table 1**. The duration of light exposure and environmental temperatures for each study period are summarised in **Table 2.** Nutritional composition and analytical constituents of forage and/or concentrate feed are summarised in **Table 3.** Staff at each study location

**Table 2. Environmental conditions during each study.**

|  | Study 1 | Study 2 | Study 3 | Study 4 |
|---|---|---|---|---|
|  | Sept.—Nov. | Jul.—Oct. | Nov.—Mar. | Jan.—Mar. |
| Maximum Temperature (˚C) | 22.5 | 22.8 | 13.4 | 12.7 |
| Minimum Temperature (˚C) | -1.0 | 1.3 | -5.4 | -4.8 |
| Average Temperature (˚C) | 10.2 | 13.2 | 4.7 | 4.1 |
| Treatment Light Exposure (hrs) | 16–15 | 17.5–15 | 14.5–15.5 | 14.5–16 |
| Control Light Exposure (hrs) | 12–9 | 16.5–10 | 7.5–11 | 8–12 |

Light exposure values reflect the increase or decrease in hours due to the natural changes in dawn and dusk times.

**Table 3. Nutritional composition and analytical constituents of feed used in each study.**

| Study | Feed type | Composition | Analytical constituents |
|---|---|---|---|
| 1 / 2 | Fibre mix | Wheatfeed, oatfeed, oatstraw, alfalfa, barley, cane molasses, extracted sunflower, calcium carbonate, maize, sodium phosphate, peas, magnesium oxide, sodium chloride, potassium chloride | Crude protein 8%, crude fibre 18%, crude oils 2%, crude ash 9% |
| | Coarse feed | Oat pellet, wheatfeed, flaked maize, rolled oats, soya hulls, molasses, soya bean meal, minerals, maize distillers meal | Crude protein 14%, crude fibre 10%, crude oil 3%, crude ash 5% |
| | Curragh Carron oil | Linseed oil, calcium hydroxide, magnesium sulphate heptahydrate | Crude protein <0.1%, crude fibre 1.2%, crude oil 41.7%, crude ash 0.2% |
| | Beet pulp | Sugar beet | Crude protein 7%, crude fibre 17.5%, crude oil 0.5%, crude ash 4.5% |
| | Hay | Unknown | Unknown |
| | Pasture | Permanent: ryegrass, clover, weeds | Unknown |
| 3 | Pasture | Permanent: ryegrass, clover, weeds | Unknown |
| | Haylage | PRG (Tyrella, Twymax, Glenariff, Soraya), HRG rusa | Unknown |
| 4 | Grass nuts | Barley, dried sugar beet pulp, soya hulls, grass meal, cane molasses, distillers dried grains, maize, soya bean meal, calcium carbonate, sodium chloride, mono dicalcium phosphate | Crude protein 12%, crude fibre 15%, crude oil 2.5%, crude ash 8.4% |
| | Pellets | Barley, wheatfeed, soya bean hulls, sugar cane molasses, oat middlings, maize, linseed, wheat, flaked peas/beans, maize gluten, distillers dried grains, calcium carbonate, fatty acids distillates, sodium chloride, soya oil blend, mono dicalcium phosphate | Crude protein 12.5%, crude fibre 10.5%, crude oil 3.75%, crude ash 6.5% |
| | Hay | Unknown | Unknown |
| | Pasture | Permanent: ryegrass, clover, weeds | Unknown |

Access to water was ad libitum in all studies.

monitored and were advised to report any perceivable changes in body condition of the experimental animals during the studies.

## Light therapy

To extend photoperiod in treatment groups, head-worn light masks (Equilume Ltd.) were used (Fig 1). Lights activated at dusk (studies 1, 3 and 4) or 8am (study 2) and deactivated at 11pm. In addition to natural daylength, this provided a photoperiod range from 14.5 to 17.5h, depending on the time of year.

## Sample collection

Hair samples (inclusive of root follicles) were collected bi-weekly from the same area below the withers on the shoulder and stored in labelled bags. Ten guard hairs (longer and thicker than under hairs) were selected for length measurement. Hairs were fixed onto a lightly coloured surface for better visibility, straightened and measured in millimetres. Average hair length was then calculated. For weight determination, ten guard hairs were weighed, using a microbalance *(Mettler Toledo UM3)*. Coat condition was scored visually by the same technician using a scale from 1 (summer coat: short, sleek and shiny) to 5 (winter coat: long, thick and dull) at the time of hair sampling. Rate of shedding was determined by the same technician in studies 3 and 4 by assigning scores to the amount of hair adhering to the hand after stroking three times in the direction of hair growth (1 no hair adhering to the hand; 2 a few hairs adhering to the hand; 3 plenty of hairs adhering to the hand; 4 plenty of hairs adhering to hand with additional hairs falling to the ground).

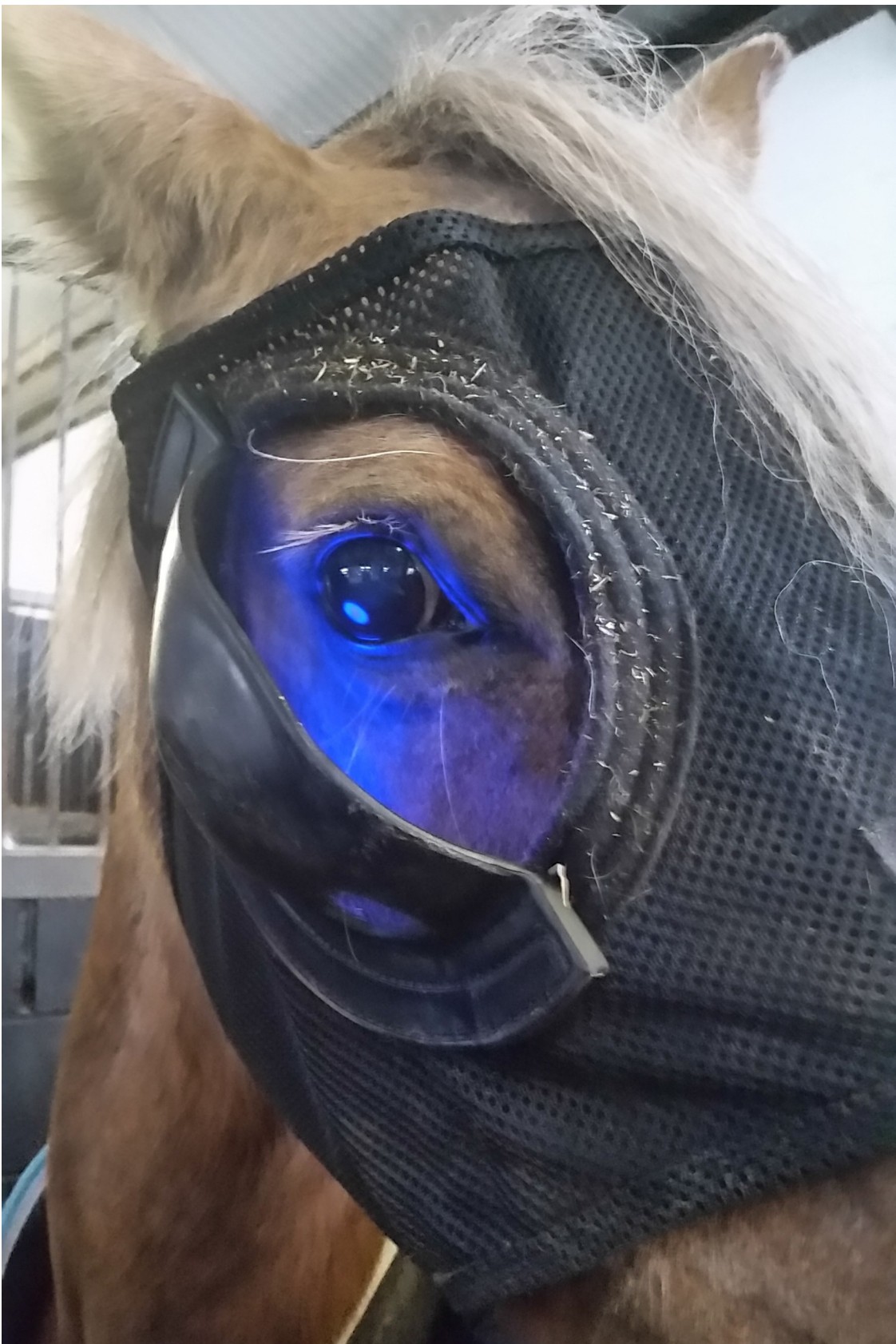

**Fig 1. Horse wearing a mobile light mask.**

## Experimental design

**Study 1:** This study was conducted from September 21st to November 8th, 2016 and used fourteen adult horses (n = 7) and ponies (n = 7) (mares and geldings) at a riding centre in County Dublin, Ireland. Animals comprised mixed breed Irish sport horses, Welsh and Shetland ponies. Both horses and ponies were used in this study to investigate potential type-related differences in treatment response. Animals were maintained in individual stables with 2-4h turn-out time each day. From October 5th onwards, animals were fitted with rugs at night to compensate for cooler temperatures. Daytime average light intensity at eye level in stables was 71.1 ± 38.6 lux (ISO-TECH ILM-01 light metre) and was provided naturally by daylight entering through windows and doorways. Overhead fluorescent lighting was seldom used during the day and never after 5pm. Animals were fed once daily with 'Classic Fibre Mix' (Dodson & Horrell, Kettering, UK) at 150g per 100kg bodyweight and 'Four Seasons Horse & Pony' (Quinns of Baltinglass Ltd., Baltinglass Ireland) coarse feed at 750g per 100kg bodyweight. Additionally, horses and ponies received 60mL and 40mL Curragh Carron Oil, plus 400g and 200g beet pulp, respectively. Hay was fed twice daily and totalled 10.5kg for horses and 7kg for ponies. Access to water was *ad libitum*. Animals grazed permanent pasture, which included a mix of ryegrass, clover and various weed species, for 2-4h per day. Quantities fed reflected individual requirements depending on workload, as determined by the yard's management. The treatment group consisted of two geldings (horses) and five mares (two horses, three ponies) (n = 7) and received daily extended photoperiod via light masks from dusk until 11pm. The control group consisted of five geldings (three horses, two ponies) and two mares (ponies) (n = 7) and received no light therapy.

**Study 2:** This study was conducted from July 21st to October 26th, 2017 at the same location as study 1 and used twelve adult horses (n = 6) and ponies (n = 6). From September 19th onwards, animals were fitted with rugs at night to compensate for cooler temperatures. The treatment group consisted of four geldings (two horses, two ponies) and two mares (one horse, one pony) (n = 6) and received daily artificial light via light masks from 8am until 11pm. The control group consisted of four geldings (three horses, one pony) and two mares (pony) (n = 6) and received no light therapy.

**Study 3:** This study was conducted from November 28th, 2017 to March 4th, 2018 and used ten Irish Sport Horses accustomed to year-round outdoor maintenance at a Sport Horse breeding farm in County Carlow, Ireland. Animals were maintained at pasture 24/7 and wore rugs for the duration of the study. They had *ad libitum* access to haylage and water in addition to grazing from permanent pasture, which included a mix of ryegrass, clover and various weed species. The treatment group consisted of three pregnant mares with due dates in April, May and June, one filly (female horse <4 years) and one gelding (n = 5) and received daily artificial light via light masks from dusk until 11pm. The control group consisted of three pregnant mares with due dates in April, May and June, one filly and one gelding (n = 5) and received no light therapy.

**Study 4:** This study was conducted from January 20th to March 19th, 2018 and used fourteen Connemara ponies at a stud in County Meath, Ireland. Except for four stallions, all ponies were maintained outdoors at pasture for the duration of the study. The stabled stallions received 1.5kg of grass nuts (Louis A. McAuley Ltd., Navan, Ireland) in the morning, 1kg of 'Pegasus Cool Mix' pellets (John Thompson & Sons Ltd., Belfast, UK) in the evening and hay twice daily, to a total of 9kg. The remaining ponies grazed permanent pasture which included a mix of ryegrass, clover and various weed species and received *ad libitum* hay. All animals had *ad libitum* access to water. No artificial light was used in stallion stables. The treatment group, consisting of two pregnant mares with due dates in March and April, one non-pregnant mare,

two fillies and two stallions (n = 7), received daily artificial light via light masks from dusk until 11pm and wore rugs. The control group consisted of two pregnant mares with due dates in March, one non-pregnant mare, two fillies and two stallions (n = 7) and did not receive artificial light or rugs.

## Data analysis

All data were analysed using GraphPad Prism version 6.0 for Mac. Data was first assessed for normal distribution using the D'Agostino and Pearson omnibus test for normality. Changes in the continuous variables length and weight were analysed by two-way repeated measures ANOVA followed by Sidak's multiple comparisons test where appropriate. The scored variables of coat condition and coat shedding were analysed using non-parametric tests (Friedman test over time within groups, Mann-Whitney U test between groups at individual time points). All data are presented as means +/- SEM and $P<0.05$ was considered significant.

## Results

**Study 1:** There was no significant time x treatment interaction ($P>0.05$) or main effect of treatment ($P>0.05$) observed. A significant effect of time on both hair length and weight ($P<0.0001$, respectively) was seen. Mean guard hair length increased by 90.1% in the treatment group (from 13.6 to 25.8mm) and by 82.2% in the control group (from 16.6 to 30.3mm) (Fig 2A). Mean hair weight (10 hairs) increased by 101.7% (from 435.4 to 878.2μg) in treatment animals, while control animals increased by 121.9% (from 459.0 to 1018.6μg) (Fig 2B). There was no difference in response to treatment between horses and ponies ($P>0.05$). Coat condition score changed over time and followed different patterns in both groups. Scores reduced in the treatment group between October 3rd and October 28th before a slight increase on November 8th ($P<0.05$). Coat condition score remained stable in the control group between September 21st and October 14th before a steep increase on October 28th ($P<0.0001$). Coat condition differed between groups on October 28th and November 8th ($P<0.01$ and $P<0.05$, respectively) where it was observed as more winter-like in appearance in the control group earlier than in the treatment group (Fig 2C). There were no reports of individual body condition changes during the study.

**Study 2:** There was an overall time x treatment effect for hair length ($P<0.01$) and hair weight ($P<0.0001$) across all animals. Mean guard hair length increased by 137.2% in the treatment group (from 8.9 to 21.2mm) and by 158.5% in the control group (from 10.6 to 27.3mm) between the beginning and end of the study period. Mean hair weight increased by 114.6% (from 254.7 to 546.6μg) in treatment animals, while control animals increased by 130.5% (from 416.9 to 961.1μg). However, differences were observed between horses and ponies (Fig 3A, 3B and 3D). In ponies, there was a significant time x treatment effect for hair weight ($P<0.05$) but not for hair length. Weight was higher in the control group than in the treatment group on October 13th ($P<0.01$). There was a strong main effect of time ($P<0.0001$) on hair length in ponies with both groups showing increases over the study period. Horses demonstrated a significant time x treatment interaction for both length and weight ($P<0.0001$ and $P<0.001$, respectively). Hair from control horses was longer than treatment horses on September 14th and 29th and on October 13th and 26th ($P<0.05$, $P<0.001$, $P<0.0001$ and $P<0.0001$, respectively). Similarly, control horses had significantly higher hair weight on October 13th and 26th ($P<0.01$ and $P<0.001$, respectively). Coat condition score changed significantly over time in both groups ($P<0.0001$, respectively). However, scores increased more rapidly in the control group such that differences in scores were observed at three time points; September

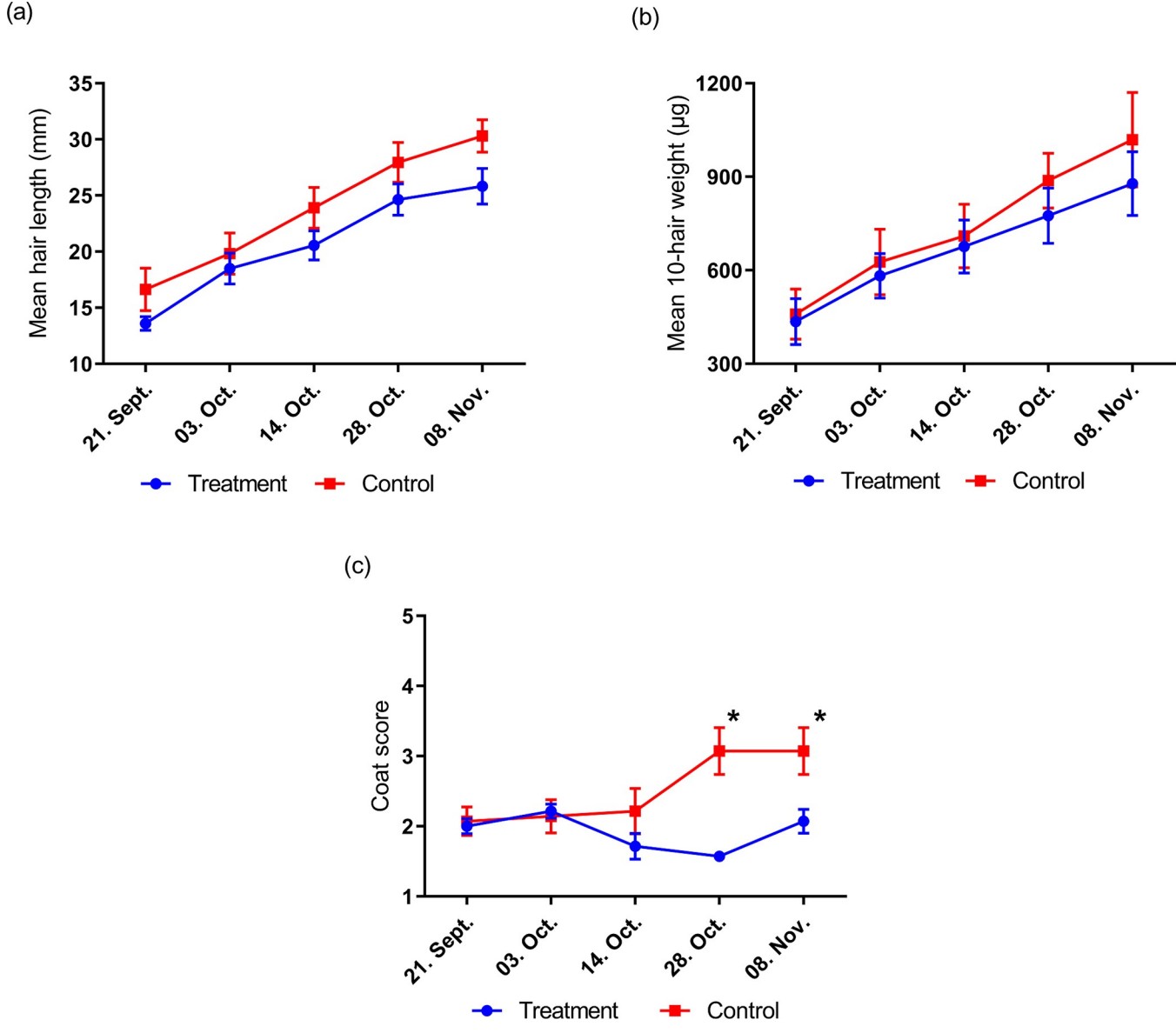

**Fig 2.** Changes in guard hair length (a), guard hair weight (b) and coat condition (c) of stabled horses and ponies in study 1. * denotes significance at P<0.05.

14th, September 29th and October 13th (P<0.05, respectively; Fig 3C). There were no reports of individual body condition changes during the study.

**Study 3:** There was a time x treatment interaction for hair length (P<0.05) and hair weight (P<0.01). No differences in length were observed on any sampling date (Fig 4A). However, significant differences in hair weight were observed between groups on February 6th and February 17th (P<0.05 and P<0.01, respectively; Fig 4B). On February 6th and February 17th, both groups displayed a notable increase in hair weight followed by a decrease on March 4th. Overall, mean guard hair length decreased by 14.4% in the treatment group (from 28.4 to 24.3mm) but rose by 31.1% in the control group (from 25.4 to 33.3mm) between the beginning and end of the study. In treatment animals, mean hair weight reduced by 17.6% (from 770.6 to 634.7μg), while control animals increased by 18.8% (from 772.3 to 917.8μg). Individual

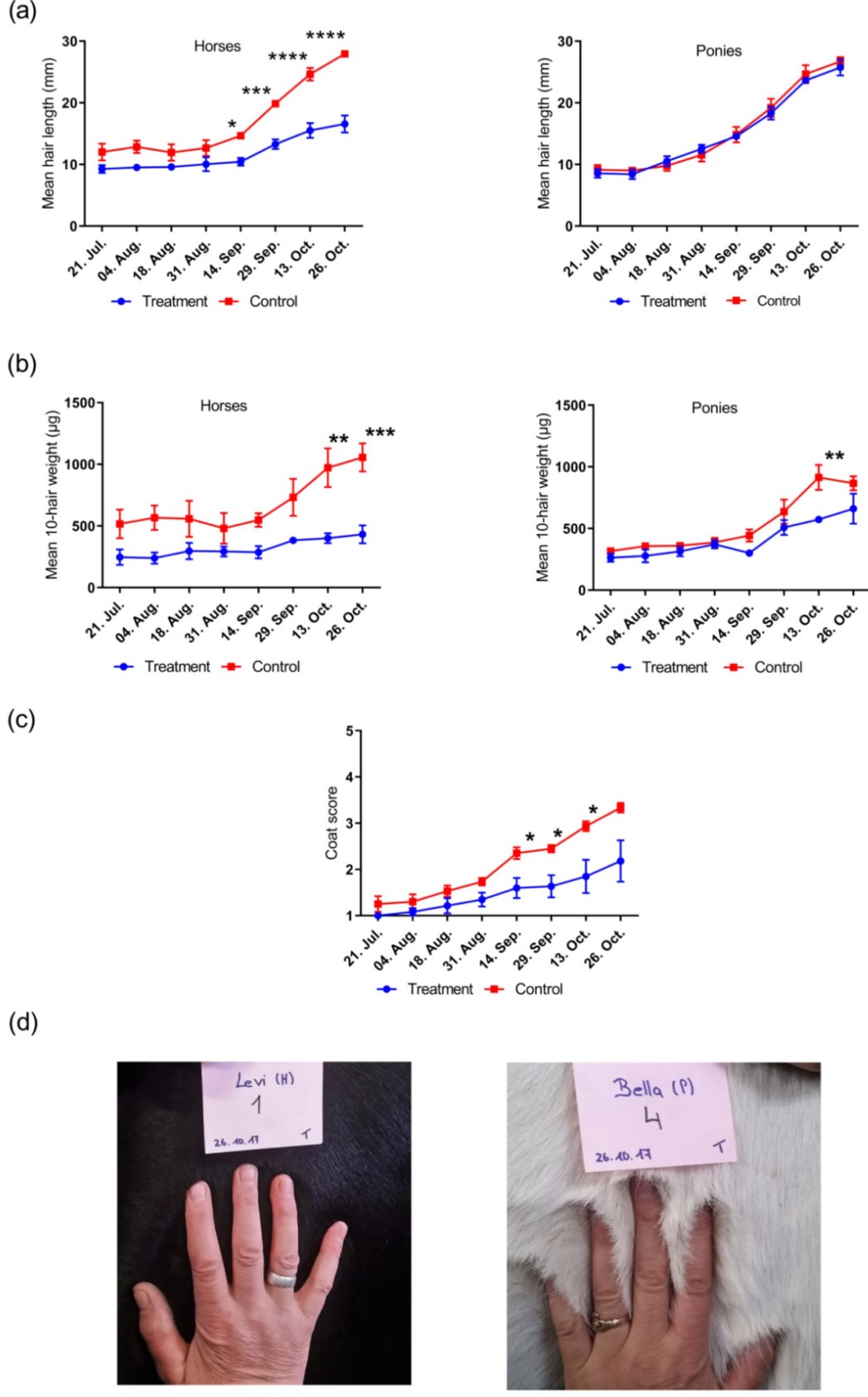

**Fig 3.** Changes in guard hair length (a), guard hair weight (b) and coat condition (c) of stabled horses and ponies in study 2. Photos taken at the end of the study period (d) depict coat differences in a treatment horse (left) and a treatment pony (right). *,**,***,**** denote significance at the P<0.05, P<0.01, P<0.001 and P<0.0001 level, respectively.

variation in response to treatment was observed. Coat condition score changed in the treatment and control group over time (P<0.05 and P<0.01, respectively; Fig 4C). Coat shedding score increased in the treatment group (P<0.01) but not in the control group (P>0.05; Fig 4D). There were no reports of individual body condition changes during the study.

**Study 4:** There was no time x treatment effect for hair length or hair weight (P>0.05). There was a significant effect of time on both variables (P<0.0001) but no effect of treatment (P>0.05), although there was a tendency towards increased length in control animals (P = 0.0711; Fig 5A and 5B). Mean guard hair length decreased by 35.2% in the treatment group (from 30.4 to 19.7 mm) and by 18.9% in the control group (from 33.4 to 27.1mm). Treatment animals experienced a reduction in mean hair weight by 65.2% (from 1407.6 to

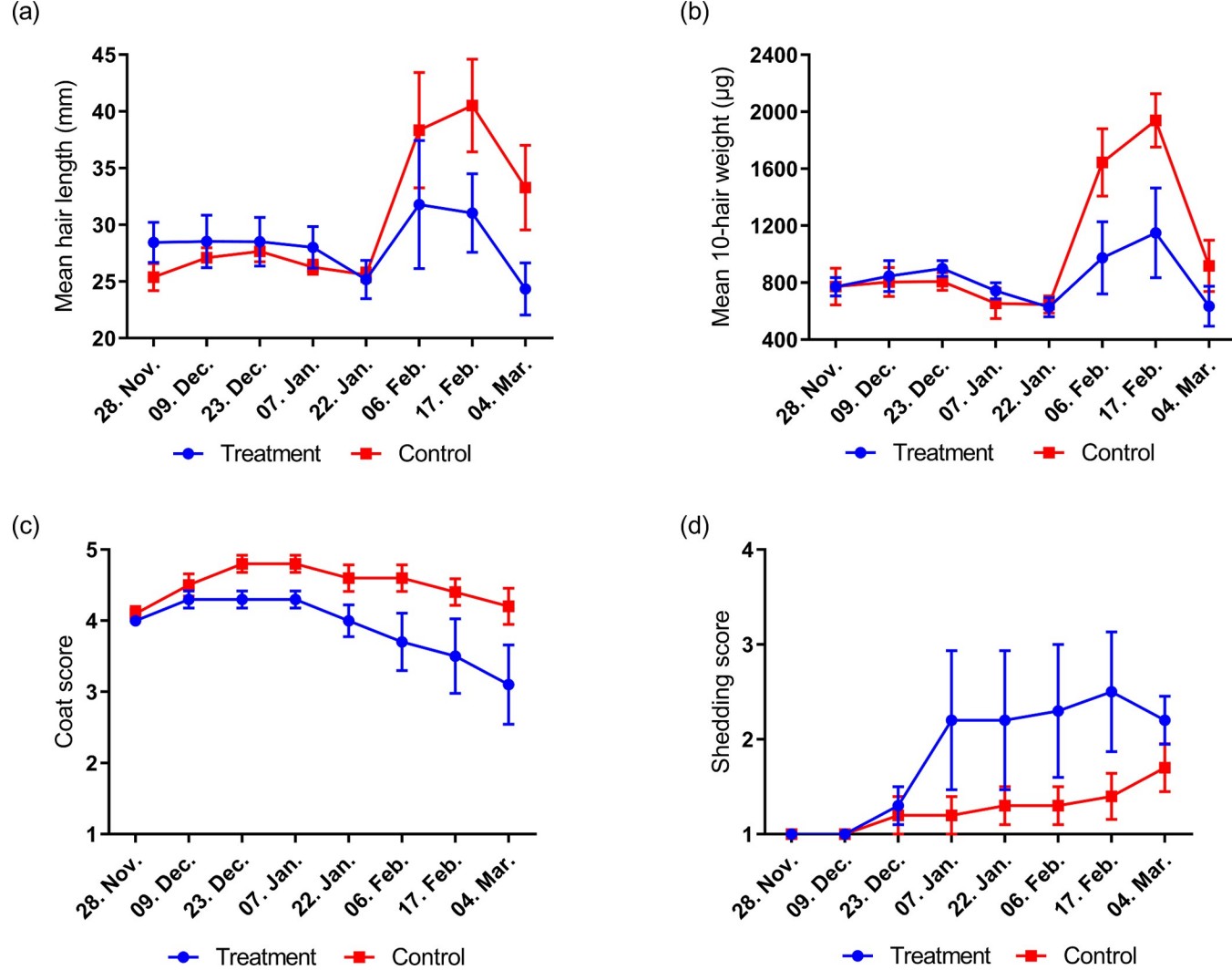

**Fig 4.** Changes in guard hair length (a), guard hair weight (b), coat condition (c) and coat shedding (d) of outdoor living horses in study 3.

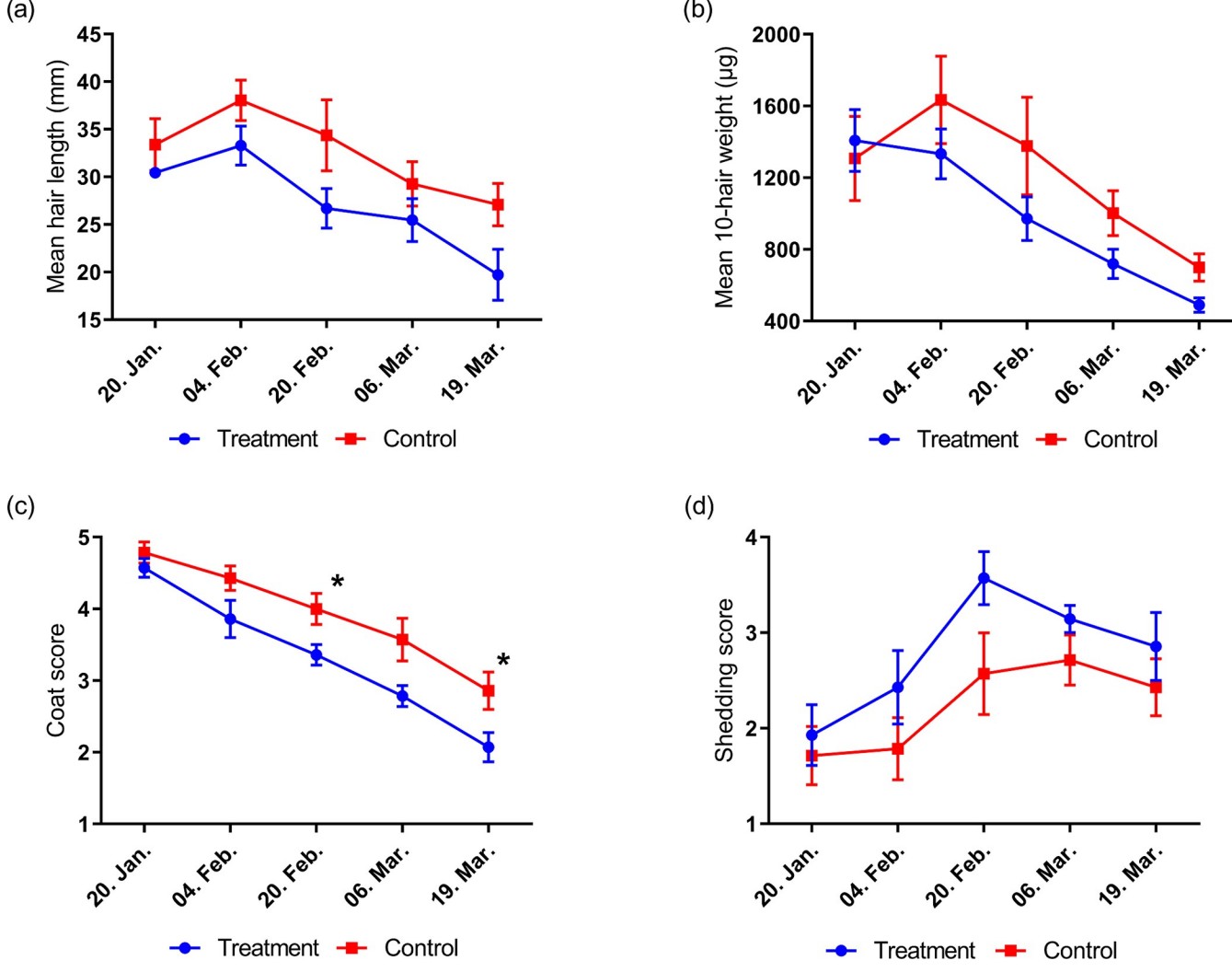

**Fig 5.** Changes in guard hair length (a), guard hair weight (b), coat condition (c) and coat shedding (d) of outdoor and indoor living ponies in study 4. * denote significance at the P<0.05 level.

489.4μg), while control animals decreased by 46.5% (from 1306.7 to 699.1μg). Coat condition score reduced significantly in both groups (P<0.0001, respectively; Fig 5C). Shedding score increased in the treatment group (P<0.05) but not in the control group (P>0.05; Fig 5D). Coat condition differed between groups on February 20th and March 19th (P<0.05, respectively) where it was observed as more summer-like in appearance in the treatment group earlier than in the control group. There were no reports of individual body condition changes during the study.

## Discussion

### Effectiveness of extending photoperiod at the autumnal equinox

Administration of extended photoperiod and warmth, when commenced at the autumnal equinox, did not reverse the early-stage winter coat growth in stabled ponies or horses. Hairs continued to increase in length and weight, which resulted in a long, thick coat by early November. In sheep in which the rate of wool growth is seasonally regulated, changes in pelage

are regulated during the transition from telogen (quiescence) to anagen (active growth) [25]. Recent interrogations of growth stages in mammalian hair follicles have identified the existence of two sub-phases of the telogen stage [11]. The authors reported that catagen (involution) is followed directly by an early, 'refractory' telogen phase, while at the end of telogen a 'competent' phase prepares follicles for transition into anagen. This supports the argument that responsiveness of hair follicles to growth stimuli changes over time. For example, A study by Plikus et al [26] found that hairs plucked from mice during early telogen took longer to regrow than those plucked during late telogen. As hair growth increased at an equal rate over time in both groups in study 1, hair follicles had already progressed to the anagen phase prior to commencement of the study and therefore were not in the suitable state (the competent stage of telogen [26]) to be regulated by the external stimulus of extended photoperiod. This is further supported by the seasonal decline in circulating prolactin levels in horses, which begins in August and falls to only two-thirds of peak production levels by mid-Sept [5] in line with decreasing daylength. Importantly, the findings of this study highlight that waiting to initiate light therapy until mid- to late September to ensure a short, sleek coat for a late autumn or winter competition season will be ineffective in both horses and ponies.

## Effectiveness of extending photoperiod one month after the summer solstice

Extending photoperiod one month after the summer solstice, before any observable changes in pelage had occurred, effectively maintained the summer coat in stabled horses, but not in stabled ponies. After 14 weeks, the hairs of treatment horses were significantly shorter and lighter compared to those of control horses. However, the hairs of all ponies grew longer, resulting in the formation of an early-stage winter coat.

As the main function of the winter coat is to insulate the body during cold climatic conditions, it is logical that temperature is an important modulator of photoperiodic influence on coat growth and was shown to be so in horses [14]. It is unlikely that growth will be delayed, or shedding accelerated, if no other mechanism, such as increased energy intake or external provision of warmth is in place to maintain core body temperature. The lower critical temperature (LCT) is a limit below which animals are required to further increase heat generation [27]. Large breeds lose less heat compared to small breeds due to a smaller heat-dissipating to heat-retaining body surface ratio [28]. This means that the LCT of horses with intact winter coats can vary between breeds and management regimes. Morgan et al. [28] estimated that ponies have an LCT of 6.9˚C when fed for maintenance and 1.4˚C when fed a higher energy diet for competition purposes. In warmblood horses these values are 2.9˚C and -3.4˚C, respectively. All animals in study 2 were fed to support low to moderate levels of exercise. Local temperatures dropped on only one occasion (mid-October) below the estimated LCT for ponies receiving a high-energy diet. However, temperatures fell below the LCT for ponies on a maintenance diet on several occasions prior to commencement of light treatment and before rugs were fitted at night. In contrast, temperatures only dropped below the maintenance LCT for larger horses [28] on two occasions (mid-October) and never below that of horses fed for competition. Additionally, a 45-minute reduction in natural daylength occurred between the summer solstice and July 21[st]. The combination of this small reduction in daylength and the occurrence of temperatures below the maintenance LCT before the study commenced, may have influenced the early transition of pony hair follicles to the autumn anagen phase and the initiation of winter coat growth. In contrast, the small reduction in daylength and temperatures above the LCT for horses, may not have been sufficient to trigger the same response in horse hair follicles. They potentially remained in the late or 'competent' summer telogen phase, during

which, according to Geyfman et al. [11] they are more responsive to stimuli such as extended photoperiod.

Seasonal adaptations, such as alterations to pelage, take time and must therefore be initiated prior to environmental conditions changing [29]. Different horse breeds are equipped with different thermoregulation strategies, including differences in behaviour, metabolism and coat [30]. As ponies are hardy, indigenous animals that are believed to have been domesticated more recently than other horse types [31], they could be more sensitive to minor changes in daylength and temperature. Another theory is that the rusticity of the pony breeds is such that they possess stronger circannual hair growth rhythms that are less amenable to phase resetting by environmental cues. The findings of this study suggest that for ponies, photoperiod may need to be maintained from the time of the summer solstice and that ambient temperatures require careful monitoring to prevent the transition to winter coat growth and to maintain the summer coat.

## Effectiveness of extending photoperiod one month before the winter solstice

Administering extended photoperiod and warmth to outdoor living horses between late November and early March, resulted in shorter and lighter hair coats in treatment horses. However, differences between groups only began to emerge a few weeks after the winter solstice.

During the first weeks, hair follicles were potentially in the refractory telogen phase and did not respond significantly to the extension of photoperiod or fluctuations in ambient temperatures. Nightly temperatures frequently dropped below 2.9°C, which is the estimated LCT for horses with intact winter coats on a maintenance diet [28]. However, no changes in hair coat occurred until after January 7th when some treatment horses started shedding. Therefore, it could be hypothesised that hair follicle cells remained unresponsive until after the winter solstice and subsequently activated follicles reacted quickly to the low environmental temperature. This may explain the sudden spike in both hair length and weight observed in both groups between January 22nd and February 17th. While there were no dramatic changes in temperatures recorded around this time, there was an added wind-chill caused by stronger winds that may have elicited a reactive growth response, however this is speculative at best. In treatment animals, this spike was less pronounced, suggesting that the light stimulus dampened this response.

During the last two weeks (mid-late February), daily average temperatures rose above 10°C for the first time. Daylength for control horses had increased by approximately 3.5h from that at the winter solstice. The combined effect on prolactin secretion of longer daylength and warmer temperatures [32,33] may have contributed to the observed decreases in hair length and weight in both groups at the end of the study period.

Individual variations in response to treatment were also observed. In the treatment group, two of the pregnant mares showed length reductions of between 25 and 38%, the third pregnant mare only showed a reduction of 5% whereas the gelding increased length by 20%. Similar variation was seen for hair weight. Responses to photoperiod can vary between populations and between individuals from the same population, potentially as a result of variation in melatonin responsiveness or its secretion pattern [34]. While previous studies have shown that melatonin production in horses is acutely suppressed by light [17], significant individual variation in night-time production levels were observed [35]. Additionally, different thermoregulatory requirements may cause hair growth and shedding to vary according to age, body region, breed and sex [36–38].

The efficacy of light treatment on the shedding of the winter coat may also differ significantly between outdoor and indoor housing systems due to the differing demands on thermoregulation. Heat loss per square meter of body surface area is estimated to be 20% greater in horses maintained outdoors compared to a four-walled shelter [39]. Stachurska et al. [38] found that air temperature significantly affected coat changes in outdoor living Polish Konik horses. The authors suggest that precipitation also contributes to a delay or slowing in shedding. Wind is another factor that is known to affect heat loss [28,30]. It is hard to evaluate the efficacy of the rugs used at protecting the study animals from heat loss. As piloerection (the ability of hairs to be raised, lowered or turned by hair erector muscles) can increase coat depth by 10 to 30%, it improves insulation [40]. It could be argued that rugs reduced this effect by flattening the hairs and thereby did not provide a sufficient substitute for the winter coat, potentially explaining slowness to shed despite light treatment when horses are maintained outdoors.

Similarly, nutritional factors may have impacted the effectiveness of light treatment in the outdoor living horses, as they were only provided with supplemental forage for basic maintenance. Although not perceived by staff managing the animals, this could have resulted in fat reserves being drawn upon, reducing a natural means of thermoregulation. Rugs may have only partially compensated for this. A limitation of this study may have been the failure to measure and better quantify small changes in body condition score during the study period. In conjunction with body condition scoring, future studies could look to include measurements of rump fat thickness to calculate total fat percentage, as a means of better quantifying natural insulation by subcutaneous fat.

As large numbers of breeding stock, particularly Thoroughbreds, are exposed to artificial photoperiod early in the year (to advance seasonal reproductive activity) and this has a significant effect on coat growth, a potential welfare concern worth highlighting is the turnout of broodmares on cold winter mornings while they are under a lighting regime which stimulates coat loss. Similarly, exercising horses during the cold winter months when they have a heavy winter coat may result in a significant chill factor during cool down periods following heavy sweating. Improved understanding from this and other studies investigating changes in coat growth by photoperiod manipulation at specific times of year should help better inform management regimes and reduce these health and welfare risks.

## Effectiveness of administering extended light on month after the winter solstice

Administration of extended photoperiod and warmth one month after the winter solstice did not significantly decrease hair length or hair weight in indoor or outdoor living Connemara ponies. While the coat of treatment animals appeared more summer-like in condition, as determined by coat scores, and shedding of the winter coat was determined to be faster in light-treated ponies, the failure of hair parameters to differ between groups was unexpected.

It is a common observation that ponies typically grow longer, heavier and denser winter coats than horses. A likely reason for this is the fact that smaller horse breeds have a larger heat-dissipating to heat-retaining body surface ratio [28] and a heavier coat reduces heat loss in colder weather. It would therefore make sense that compared to larger breeds, winter coat growth in ponies is initiated earlier, as supported by observations in study 2. Similarly, this hypothesis may explain the reduced effectiveness of light treatment on hair changes in this study. Hair follicle transition to the active anagen phase may have been slowed by temperatures below the LCT limit, estimated to be between 1.4°C and 6.9°C [28]. As average nightly

temperatures during March were still at 0.9˚C, this may have dampened the effects of extended photoperiod on follicle response.

Schmidt et al. [14] found that guard hairs of Shetland pony stallions stabled between October and March at temperatures above +10˚C were shorter, hairs regrew more slowly and changes in coat appeared earlier compared to those stallions maintained outdoors, despite both groups experiencing the same changes in daylength. The authors concluded that environmental temperature significantly affected pony hair coats. Similarly, Schrammel et al. [41] observed earlier shedding of the winter coat in stabled Shetland pony stallions receiving 16 h of light from December 15th until March compared to control animals. Ten of the 14 Connemara ponies used in the current study were maintained outdoors at pasture, potentially explaining the contrasting results between studies using similarly rustic pony breeds. In the present study, shedding rate increased in both groups from February onwards but was significant only in the light-treated group.

To accelerate shedding of the winter coat in ponies effectively, it may therefore be particularly important that light treatment is combined with temperatures above 1.4˚C, when feeding a high-energy diet, or above 6.9˚C when feeding for maintenance. Frequent grooming may also affect the timing of shedding as it is thought to help to loosen hairs [38]. Ponies in this study were not groomed.

Advancing the transition to a short summer coat is desirable in many show breeds and is commonly achieved in competition show barns where ponies are maintained indoors, rugged, fed a high-energy diet, regularly groomed and exposed to 16 h artificial light/day. In this context, failure to achieve comparable results may be explained whereby extended daily light was administered to a cohort of an indigenous pony breed, primarily maintained outdoors and fed a maintenance diet.

## Conclusion

We conclude that the timing of the initiation of light therapy for modulation of coat growth greatly influences the outcome. Individual differences in response to treatment can occur and ponies respond differently to horses, potentially requiring earlier light application at the time of the solstices. While photoperiod seems to be the primary environmental stimulus regulating pelage dynamics, there are a number of factors that clearly work in concert with light to modulate the response. Different climatic conditions, thermoregulatory requirements and feeding regimes in indoor and outdoor maintenance systems may play important roles. Temperature appears to be a particularly important modulator of coat growth in ponies. A significant limitation of the current studies was the inability to control for ambient temperature conditions and follow-up investigations should rigorously control for this important environmental factor. This could be achieved by housing animals in temperature-controlled stabling at night and ensuring adequate blanketing during turn-out time. Furthermore, the hair cycle phases may determine how well follicles respond to the stimulus of extended photoperiod. To inform on a best practise approach for light-therapy-assisted coat manipulation, future studies should make special considerations for the potential influence of breed, management and hair cycle phase at initiation of treatment.

## Supporting information

**S1 Tables. Study 1 dataset.** Data include hair length, hair weight and coat scores for all individuals from study 1.
(XLSX)

**S2 Tables. Study 2 dataset.** Data include hair length, hair weight and coat scores for all individuals from study 2.
(XLSX)

**S3 Tables. Study 3 dataset.** Data include hair length, hair weight, coat scores and shedding scores for all individuals from study 3.
(XLSX)

**S4 Tables. Study 4 dataset.** Data include hair length, hair weight, coat scores and shedding scores for all individuals from study 4.
(XLSX)

## Acknowledgments

The authors gratefully acknowledge Aoife and Gerry Osborne (Osborne Farm, Co. Carlow), Ann Keogan (Fettercairn Youth Horse Project, Co. Dublin) and Mick McMenamon (Newgrange Stud, Co. Meath) for providing the horses and ponies involved in these studies. Further thanks to Kathy Heaney and Donal O'Brien for their help with fitting masks and rugs, as well as their assistance during sampling. Thanks to Anne Connolly (University College Dublin) for her help facilitating the weighing of samples.

## Author Contributions

**Conceptualization:** Barbara Anne Murphy.

**Data curation:** Christiane O'Brien, Megan Ruth Darcy-Dunne, Barbara Anne Murphy.

**Formal analysis:** Christiane O'Brien, Barbara Anne Murphy.

**Funding acquisition:** Barbara Anne Murphy.

**Investigation:** Christiane O'Brien, Barbara Anne Murphy.

**Methodology:** Christiane O'Brien, Megan Ruth Darcy-Dunne.

**Project administration:** Christiane O'Brien.

**Supervision:** Barbara Anne Murphy.

**Writing – original draft:** Christiane O'Brien, Barbara Anne Murphy.

**Writing – review & editing:** Christiane O'Brien, Megan Ruth Darcy-Dunne, Barbara Anne Murphy.

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
