## [Decision Letter · Decision Letter 0]

2 Oct 2019

PONE-D-19-23480

The effects of extended photoperiod and warmth on hair growth in ponies and horses at different times of year

PLOS ONE

Dear Mrs O'Brien,

Thank you for submitting your manuscript to PLOS ONE. After careful consideration, we feel that it has merit but does not fully meet PLOS ONE’s publication criteria as it currently stands. Therefore, we invite you to submit a revised version of the manuscript that addresses the points raised during the review process.We would appreciate receiving your revised manuscript by Nov 16 2019 11:59PM. To enhance the reproducibility of your results, we recommend that if applicable you deposit your laboratory protocols in protocols.io, where a protocol can be assigned its own identifier (DOI) such that it can be cited independently in the future. For instructions see: http://journals.plos.org/plosone/s/submission-guidelines#loc-laboratory-protocols

We look forward to receiving your revised manuscript.

Kind regards,

Paul A. Bartell

Academic Editor

PLOS ONE

Journal Requirements:

2.  Thank you for stating the following in the Competing Interests:

'I have read the journal's policy and the authors of this manuscript have the following

competing interests: BAM is a member of the Board of Directors for Equilume Ltd. and COB is an employee

of Equilume Ltd.'

We note that one or more of the authors have an affiliation to the commercial funders of this research study : Equilume Ltd

'These studies were part funded by Equilume Ltd. and the School of Agriculture and Food Science, University College Dublin.'

'COB is an employee of Equilume Ltd (www.equilume.com) who provided the light

masks used in these studies. She was involved in data collection and analysis and in

preparation of the manuscript.'

Additional Editor Comments (if provided):

Reviewers' comments:

Reviewer's Responses to Questions

**Comments to the Author**

1. Is the manuscript technically sound, and do the data support the conclusions?

Reviewer #1: Partly

Reviewer #2: Yes

2. Has the statistical analysis been performed appropriately and rigorously? 

Reviewer #1: Yes

Reviewer #2: Yes

3. Have the authors made all data underlying the findings in their manuscript fully available?

Reviewer #1: Yes

Reviewer #2: Yes

4. Is the manuscript presented in an intelligible fashion and written in standard English?

Reviewer #1: Yes

Reviewer #2: Yes

5. Review Comments to the Author

Reviewer #1: It is known that manipulation of photoperiod can accelerate development in horses, and the authors of this study tried to determine if hair growth could also be modified by manipulation of photoperiod. The purpose of changing hair growth is to avoid the winter coat, which is undesired because it does not look as good as the summer coat and because it slows down heat loss from exercising horses.

The authors find that hair growth can be slowed down by manipulation of photoperiod (by lengthening the day with artificial lighting). Not surprisingly, manipulation of photoperiod was effective only at the right time of the year (namely, shortly after the summer solstice but not after the autumnal equinox).

The Introduction is clear and adequate. The Methods are appropriate, except perhaps that the procedure could be tested more systematically over the year. The extension of photoperiod could start at various times of the year and last for different amounts of time for a more thorough examination of the process.

The Discussion has a little bit too much speculation about the role of ambient temperature. Because ambient temperature was not experimentally adjusted, the role of ambient temperature cannot be determined with any certainty. Placing a blanket on the horse's back most likely reduced cold stress on cold days, but the authors provide no objective measurement of thermal protection. A follow-up study should rigorously control ambient temperature, and this limitation of the current study should be pointed out.

To extend the photoperiod, the authors used a commercial blue-light stimulator next to the horse's eye. This stimulation worked, but the authors should point out that there was nothing special about this stimulator. For animals housed in stables, light fixtures on the ceiling would work just as well. Importantly, because spectral sensitivity follows an inverted U-shaped curve, almost any color within the visible range would likely work if presented with higher illuminance.

Reviewer #2: Overall, a well written and interesting paper. A few comments/questions that I would appreciate the authors addressing, both here and within the paper as necessary.

1. How does age impact the hair growth cycle? Particularly for study one, where there was a significant difference in age between the groups. It is mentioned down in the results (line 415) that age can cause hair growth to vary, but how much exactly?

2. In the intro it is mentioned that short wavelength blue light suppresses melatonin production in horses - has this been documented in other species?

3. Did the same person assign scores for rate of shedding, or was it multiple people?

4. What was the average weight for the horses and ponies? Would that potentially impact the rate of coat growth? It is noted that the lack of measurement of BCS is a limitation, and I would agree.

5. For the 4 studies, male horses are described as either male, gelding, or stallion. Please be consistent and refer to the males as either geldings or stallions.

6. What were the weights of the rugs used? Would the weights have an impact on results? Were all rugs within a study the same weight?

7. for the study that looked at extending the photoperiod one month before the winter solstice, it is mentioned that one gelding increased length in the treatment group. Why might this have happened, beyond just "individual variation"? Has this happened before in other horses?

8. It is mentioned briefly in the intro that knowing the effect of artificial light could improve health and welfare in breeding stock. But this is not touched upon in the discussion. it should be brought up again with some discussion. There were broodmares in one study - was anything health/welfare related noticed during the study?

6. PLOS authors have the option to publish the peer review history of their article (what does this mean?). If published, this will include your full peer review and any attached files.

Reviewer #1: No

Reviewer #2: No

---

## [Author Response · Author response to Decision Letter 0]

14 Nov 2019

Response to Reviewers: 

Reviewer #1: 

Comment 1: It is known that manipulation of photoperiod can accelerate development in horses, and the authors of this study tried to determine if hair growth could also be modified by manipulation of photoperiod. The purpose of changing hair growth is to avoid the winter coat, which is undesired because it does not look as good as the summer coat and because it slows down heat loss from exercising horses. 

The authors find that hair growth can be slowed down by manipulation of photoperiod (by lengthening the day with artificial lighting). Not surprisingly, manipulation of photoperiod was effective only at the right time of the year (namely, shortly after the summer solstice but not after the autumnal equinox). 

The Introduction is clear and adequate. The Methods are appropriate, except perhaps that the procedure could be tested more systematically over the year. The extension of photoperiod could start at various times of the year and last for different amounts of time for a more thorough examination of the process. 

The Discussion has a little bit too much speculation about the role of ambient temperature. Because ambient temperature was not experimentally adjusted, the role of ambient temperature cannot be determined with any certainty. Placing a blanket on the horse's back most likely reduced cold stress on cold days, but the authors provide no objective measurement of thermal protection. A follow-up study should rigorously control ambient temperature, and this limitation of the current study should be pointed out. 

Response: We thank the reviewer for his/her comments. We agree that there is considerable speculation in regards to temperature, but we have ensured that the comments are supported as much as possible by related evidence from the literature and backed by logical scientific argument. 

In response to the reviewer’s point on the absence of an objective measurement of thermal protection, we have now included further details on the weight of the blankets used in the studies in the methods section. These were medium weight all weather rugs (280g polyester fill) suitable for the Irish climate and a low temperature range of -6.7 to 4.4°C and included neck covers for maximum insulation against temperature drops. While this still does not provide an accurate thermal measurement, we hope that it provides relevant additional information. 

We agree that a follow-up study should control for ambient temperature and now strongly highlight this within the conclusion as follows: 

“A significant limitation of the current studies was the inability to control for ambient temperature conditions and follow-up investigations should rigorously control for this important environmental factor. This could be achieved by housing animals in temperature-controlled stabling at night and ensuring adequate blanketing during turn-out time.” 

Comment 2: To extend the photoperiod, the authors used a commercial blue-light stimulator next to the horse's eye. This stimulation worked, but the authors should point out that there was nothing special about this stimulator. For animals housed in stables, light fixtures on the ceiling would work just as well. Importantly, because spectral sensitivity follows an inverted U-shaped curve, almost any color within the visible range would likely work if presented with higher illuminance. 

Response: Respectfully, the spectral sensitivity of intrinsically photosensitive retinal ganglion cells (ipRGCs) to incandescent, fluorescent and LED lighting varies significantly and is wavelength dependent due to the presence of the photopigment melanopsin in these unique cells with their specific non-image forming functions within the brain. It was originally believed that all light signals for image and nonimage forming functions alike began and ended with the rods and cones. The discovery of ipRGCs and their unique role in circadian regulation led to many discoveries of their novel properties as a photoreceptor, not least of all the fact that they depolarize in response to light, similar to invertebrate photoreceptors, but opposite to the hyperpolarizing response of rods and cones. These melanopsin-containing ipRGCs mediate photoentrainment of the circadian system and relay time-of-day messages to the pineal gland to modulate daily changes in melatonin secretion. Preferentially stimulated by blue light, studies have shown in multiple species that blue light rather than red light optimally suppresses melatonin production (Figueiro and Rea, 2010; Murphy et al., 2019). Thus, there is indeed something special about a blue light stimulator in that it more effectively suppresses melatonin at lower intensities than broad spectrum white light as will be explained further below. 

It is accepted that high intensity broad spectrum white light works well at supressing melatonin, because white light by definition contains a proportion of blue wavelength light. However, from the authors observations and experiences of varying stable lighting conditions, the intensity is seldom optimal and this can lead to reduced effectiveness in manipulating circannual rhythms. Similarly, the type of light source emitting the white light impacts on the level of biological effectiveness depending on the proportion of blue wavelengths emitted. Fluorescent and incandescent sources have much lower levels of blue wavelength light than LED sources (Abdel-Rahman et al., 2017). In further support of this, a recent paper evaluating stimulation of ovarian activity in mares housed under different environmental conditions confirmed that significantly improved responses occurred where mares were exposed to higher amounts of natural daylight, with its high blue component (Dini et al., 2019). As the light duration was similar across all treatment groups in the Dini (2019) study, the authors propose that it was the difference in spectral composition that impacted the outcome. Interestingly, the barn with the coldest recorded ambient temperature but the greatest exposure to natural daylight showed the greatest response. The mask employed in our study was special in the sense that it only emits short-wavelength blue light that is known to optimally stimulate ipRGCs and supress melatonin in horses (Walsh et al., 2013). In rebuttal of the reviewer’s suggestion that almost any color within the visible range would likely work if presented with higher illuminance, we would draw attention to two studies that used light masks to hasten ovulation in mares but differed only in the colour and intensity of the light source that was delivered to a single eye of the treatment mares. In Murphy et al., (2014) lengthened daily photoperiod provided by 50 lux narrow spectrum blue wavelength light initiated on Dec 1 resulted in 87.5% of mares ovulating by Feb 10th. In comparison, an earlier study also initiated on Dec 1 with light delivered to one eye but utilising a higher intensity, 130 lux white incandescent light (higher in the yellow-orange spectrum) only reported 67% mares having ovulated by April 1, seven weeks later in the season (Shabpareh et al., 1992). In this case, higher intensity white light had a significantly reduced effect at seasonally advancing reproduction in horses than lower intensity blue light. Presumably, the same would hold true for advancing seasonal moulting. 

Furthermore, the use of this device facilitated improved experimental design. In stabled conditions its use allowed the housing of treatment and control animals within the same barn / environmental conditions without the treatment light affecting control animals. Using a ceiling light would have caused light spill-over to adjacent or close proximity stalls where animals were housed under one roof. Additionally, the head-worn aspect of the device further allowed studying the effects of artificial photoperiod in animals maintained under natural outdoor conditions. 

Respectfully, as we have not in any way suggested in the manuscript that the device used was ‘special’, equally we feel justified in not highlighting that the device was not special. 

Reviewer #2: Overall, a well written and interesting paper. A few comments/questions that I would appreciate the authors addressing, both here and within the paper as necessary. 

1. How does age impact the hair growth cycle? Particularly for study one, where there was a significant difference in age between the groups. It is mentioned down in the results (line 415) that age can cause hair growth to vary, but how much exactly? 

Response: Little is known about how age influences hair growth in horses apart from the higher prevalence of pituitary pars intermedia dysfunction (PPID) or Cushing’s Disease in aged horses. A symptom of this condition is excessive hair growth (hypertrichosis) and is due to persistence of hair follicles in the anagen phase. While not specifically tested for in the study cohort, there were no overt signs that any were suffering from this condition. However, we do cite Stachurska et al. (2015) (line 419) as they commented on a more general point that differences have been observed in coat shedding of foals, old, ill, or thin horses. As neither foals nor very old, ill or thin animals were used in our studies we did not feel it necessary to expand on this point further. 

2. In the intro it is mentioned that short wavelength blue light suppresses melatonin production in horses - has this been documented in other species? 

Response: Yes, it has, most commonly in mice and humans. Please see following references which have now been added to the introduction: (Brainard et al., 2008; Thapan et al., 2001; West et al., 2011). 

3. Did the same person assign scores for rate of shedding, or was it multiple people? 

Response: Yes, the same person assigned scores for shedding across the studies. This has been clarified in the Sample collection section. 

4. What was the average weight for the horses and ponies? Would that potentially impact the rate of coat growth? It is noted that the lack of measurement of BCS is a limitation, and I would agree. 

Response: Data on the average weight for the horses and ponies was regrettably not available. Private riding schools and breeding yards in Ireland are rarely equipped with weighing scales. Weight may have been estimated using other methods but we respectfully believe that such estimates would not be sufficient to determine the effect of body weight on the rate of hair growth. The possible effect of weight and BCS on the rate of coat growth could be addressed in future studies. However, such studies should include measurements of rump fat thickness to calculate the total fat percentage, as fat may act as a natural insulator. We have further addressed this limitation in the manuscript by adding the following statement: 

“In conjunction with body condition scoring, future studies could look to include measurements of rump fat thickness to calculate total fat percentage, as a means of better quantifying natural insulation by subcutaneous fat.” 

5. For the 4 studies, male horses are described as either male, gelding, or stallion. Please be consistent and refer to the males as either geldings or stallions. 

Response: We have gone back through the manuscript and improved consistency by referring to males as either geldings or stallions. 

6. What were the weights of the rugs used? Would the weights have an impact on results? Were all rugs within a study the same weight? 

Response: All four studies used the same medium-weight rugs. These consisted of a waterproof outer layer and 280 g polyester fill, suitable for a mild climate with a low temperature range of -6.7 to 4.4°C and they included neck covers for maximum insulation against temperature drops. We have added this information to the methods section of the manuscript. 

7. for the study that looked at extending the photoperiod one month before the winter solstice, it is mentioned that one gelding increased length in the treatment group. Why might this have happened, beyond just "individual variation"? Has this happened before in other horses? 

Response: A similar increase in hair length to that observed in the study 3 treatment gelding occurred only within the study 3 control group. As treatment geldings in another study (study 2) responded well to the light stimulus, sex appears to be an unlikely cause and the reason for this particular gelding’s unresponsiveness to light treatment remains unknown. 

8. It is mentioned briefly in the intro that knowing the effect of artificial light could improve health and welfare in breeding stock. But this is not touched upon in the discussion. it should be brought up again with some discussion. There were broodmares in one study - was anything health/welfare related noticed during the study? 

Response: Due to the fact that large numbers of breeding stock are exposed to artificial photoperiod early in the year (to advance seasonal reproductive activity) and that this has a significant effect on coat growth, a potential associated welfare concern is the turnout of broodmares on cold winter mornings while they are under a lighting regime which stimulates coat loss. Similarly, exercising horses during the cold winter months when they have a heavy winter coat may result in a significant chill factor during cool down periods following heavy sweating. Improved understanding of coat manipulation by photoperiod at specific times of year should help better inform management regimes and reduce these health and welfare risks. Within the current studies there were no health/ welfare issues noticed as the animals were optimally managed for their respective indoor and outdoor environments. The above has been expanded upon within discussion at the end of the section discussing Study 3. 

  

 References cited: 

Abdel-Rahman, F., Okeremgbo, B., Alhamadah, F., Jamadar, S., Anthony, K., Saleh, M.A., 2017. Caenorhabditis elegans as a model to study the impact of exposure to light emitting diode (LED) domestic lighting. J. Environ. Sci. Heal. - Part A Toxic/Hazardous Subst. Environ. Eng. 

Brainard, G.C., Sliney, D., Hanifin, J.P., Glickman, G., Byrne, B., Greeson, J.M., Jasser, S., Gerner, E., Rollag, M.D., 2008. Sensitivity of the Human Circadian System to Short-Wavelength (420-nm) Light. J. Biol. Rhythms 23, 379–386. 

Dini, P., Ducheyne, K., Lemahieu, I., Wambacq, W., Vandaele, H., Daels, P., 2019. Effect of environmental factors and changes in the body condition score on the onset of the breeding season in mares. Reprod. Domest. Anim. 

Figueiro, M.G., Rea, M.S., 2010. The Effects of Red and Blue Lights on Circadian Variations in Cortisol, Alpha Amylase, and Melatonin. Int. J. Endocrinol. 2010, 1–9. 

Murphy, B.A., O’Brien, C., Elliott, J.A., 2019. Red light at night permits the nocturnal rise of melatonin production in horses. Vet. J. 252, 105360. 

Murphy, B.A., Walsh, C.M., Woodward, E.M., Prendergast, R.L., Ryle, J.P., Fallon, L.H., Troedsson, M.H.T., 2014. Blue light from individual light masks directed at a single eye advances the breeding season in mares. Equine Vet. J. 46. 

Shabpareh, V., Squires, E.L., Cook, V.M., Cole, R., 1992. An alternative artificial lighting regime to hasten onset of the breeding season in. Equine Pract. 14, 24–27. 

Thapan, K., Arendt, J., Skene, D.J., 2001. An action spectrum for melatonin suppression: evidence for a novel non-rod, non-cone photoreceptor system in humans. J. Physiol. 535, 261–267. 

Walsh, C.M., Prendergast, R.L., Sheridan, J.T., Murphy, B.A., 2013. Blue light from light-emitting diodes directed at a single eye elicits a dose-dependent suppression of melatonin in horses. Vet. J. 196, 231–235. 

West, K.E., Jablonski, M.R., Warfield, B., Cecil, K.S., James, M., Ayers, M.A., Maida, J., Bowen, C., Sliney, D.H., Rollag, M.D., Hanifin, J.P., Brainard, G.C., 2011. Blue light from light-emitting diodes elicits a dose-dependent suppression of melatonin in humans. J. Appl. Physiol. 110, 619–626.

---

## [Editor Report · Decision Letter 1]

13 Dec 2019

The effects of extended photoperiod and warmth on hair growth in ponies and horses at different times of year

PONE-D-19-23480R1

Dear Dr. O'Brien,

We are pleased to inform you that your manuscript has been judged scientifically suitable for publication and will be formally accepted for publication once it complies with all outstanding technical requirements.

With kind regards,

Paul A. Bartell

Academic Editor

PLOS ONE
---

## [Editor Report · Acceptance letter]

3 Jan 2020

PONE-D-19-23480R1 

The effects of extended photoperiod and warmth on hair growth in ponies and horses at different times of year 

Dear Dr. O'Brien:

I am pleased to inform you that your manuscript has been deemed suitable for publication in PLOS ONE. Congratulations! Your manuscript is now with our production department. 

With kind regards,

on behalf of

Dr. Paul A. Bartell 

Academic Editor

PLOS ONE